# A Computational Approach to Identification of Candidate Biomarkers in High-Dimensional Molecular Data

**DOI:** 10.3390/diagnostics12081997

**Published:** 2022-08-18

**Authors:** Justin Gerolami, Justin Jong Mun Wong, Ricky Zhang, Tong Chen, Tashifa Imtiaz, Miranda Smith, Tamara Jamaspishvili, Madhuri Koti, Janice Irene Glasgow, Parvin Mousavi, Neil Renwick, Kathrin Tyryshkin

**Affiliations:** 1School of Computing, Queen’s University, Kingston, ON K7L 3N6, Canada; 2Department of Pathology and Molecular Medicine, Queen’s University, Kingston, ON K7L 3N6, Canada; 3Department of Pathology & Laboratory Medicine, SUNY Upstate Medical University, Syracuse, NY 13210, USA; 4Department of Biomedical and Molecular Sciences, Queen’s University, Kingston, ON K7L 3N6, Canada

**Keywords:** biomarker, feature selection, big data analysis, RNA-Seq, prostate adenocarcinoma

## Abstract

Complex high-dimensional datasets that are challenging to analyze are frequently produced through ‘-omics’ profiling. Typically, these datasets contain more genomic features than samples, limiting the use of multivariable statistical and machine learning-based approaches to analysis. Therefore, effective alternative approaches are urgently needed to identify features-of-interest in ‘-omics’ data. In this study, we present the molecular feature selection tool, a novel, ensemble-based, feature selection application for identifying candidate biomarkers in ‘-omics’ data. As proof-of-principle, we applied the molecular feature selection tool to identify a small set of immune-related genes as potential biomarkers of three prostate adenocarcinoma subtypes. Furthermore, we tested the selected genes in a model to classify the three subtypes and compared the results to models built using all genes and all differentially expressed genes. Genes identified with the molecular feature selection tool performed better than the other models in this study in all comparison metrics: accuracy, precision, recall, and F1-score using a significantly smaller set of genes. In addition, we developed a simple graphical user interface for the molecular feature selection tool, which is available for free download. This user-friendly interface is a valuable tool for the identification of potential biomarkers in gene expression datasets and is an asset for biomarker discovery studies.

## 1. Introduction

High-throughput molecular profiling of clinical or biological samples creates unparalleled opportunities for pattern recognition and advanced data exploration. Despite advances in machine learning methods, few are utilized in the analysis of biomedical data. Currently, the most common approach to identify genes of interest or other molecular features in large datasets is to perform a differential expression analysis where each feature is assessed individually using statistical tests or curve fitting [1,2]. However, simply relying on the *p*-value for identifying features can be misleading, as the *p*-value is confounded by its dependence on sample size [3,4,5]. Thus, there is a need for more stable approach to detect patterns and interactions in -omics datasets for biomarker discovery by utilizing the high-dimensionality (number of features) of the data.

Feature selection is a machine learning technique for identifying features of importance in high dimensional data. The main goal of the feature selection is to reduce the number of features to a small set of most variable predictors required for classification. A typical -omics dataset has a relatively low sample size compared to the number of features. This affects classifier accuracy and increases the risk of overfitting the prediction model by developing an overly complex classifier with perfect classification on a training set, but low performance and generalizability on unseen data [6,7,8,9,10,11,12,13]. Therefore, when developing generalizable prediction models from -omics data, it is favorable to use only the most important predictors. Feature selection is a supervised learning approach that evaluates features either individually (univariable approach) or as a group (multivariable approach). Feature selection algorithms are typically based on (i) filter methods that evaluate each feature without any learning involved; (ii) wrapper methods that use machine learning techniques for identifying features of importance; or (iii) embedded methods where the feature selection is embedded with the classifier construction [6].

Many of the proposed applications of feature selection in -omics studies use only one type of feature selection algorithm [14,15,16,17,18,19]. However, no algorithm is superior or inferior overall according to the ‘no free lunch theorem’ [7]. Therefore, a feature selection that uses only one method may work for some datasets, but not others. It is recognized that an ensemble feature selection provides a more generalizable approach to identify features of importance [20]. While some studies use an ensemble technique [21,22], they only use a small set of either filter based methods (n = 5 algorithms [21]) or wrapper/embedded methods (n = 4 algorithms [22]); others use bootstrap aggregating [7] to perform multiple iterations of the same feature selection algorithm on different subsets of data [16,17,18]. To date, no feature selection approach for identifying biomarkers that combines multiple univariable and multivariable filter-, wrapper-, and embedded-type feature selection algorithms has been proposed and tested on multiple -omics datasets.

In this manuscript, we present our *Molecular Feature Selection Tool (MFeaST)* that enables the identification of candidate biomarkers in -omics data by the reduction of features to the most valuable predictors. This novel, ensemble-type feature selection technique does not rely just on one approach, but utilizes multiple univariable and multivariable as well as filter-, wrapper-, and embedded-type feature selection techniques. The algorithm applies a greedy method [23] to rank all available features based on the ensemble results of a diverse selection of methods and families of predictors. To date, we have successfully applied *MFeaST* to identify features of interest in many different -omics datasets [24,25,26,27,28,29,30,31,32,33,34]. Here, we demonstrate the effectiveness of *MFeaST* in selecting a small set of features as potential biomarkers in prostate adenocarcinoma mRNA profiles using immune-function related genes. Our analysis showed that the biomarkers selected using *MFeaST* could also be used to build classification models that perform better in all comparisons than the other models built in this study using differentially expressed genes or without feature selection.

## 2. Materials and Methods

### 2.1. Feature Selection Algorithm

We developed *the Molecular Feature Selection Tool (MFeaST)* to rapidly identify candidate biomarker molecules in -omics data using an ensemble feature selection approach. This MATLAB-based application accepts a feature matrix of -omics profiles and ranks each feature based on its ability to discriminate between two classes. Features are assessed for classificatory ability using an ensemble of multiple univariable and multivariable filter-, wrapper-, and embedded-type feature selection algorithms (Table 1); many of these algorithms are implemented by default in MATLAB. Subsequently, a greedy algorithm is applied to combine the results of the different feature selection methods and to rank each feature based on the ensemble vote. Several choices for cross-validation including K-fold, hold-out, leave-one-out, and resubstitution can be selected to increase the generalizability and stability of *MFeaST* and avoid potential overfitting. In this paper, *MFeaST* was applied to prostate adenocarcinoma mRNA profiles to select features for discrimination between three molecular subtypes.

### 2.2. Data Acquisition

mRNA expression profiles (n = 550) from frozen prostate adenocarcinoma samples were obtained from the TCGA-PRAD dataset [35] from the Cancer Genome Atlas (TCGA). Briefly, RNA-Seq data were aligned by the TCGA Research Network to the GRCh37 reference genome using STAR v2.30e, generating gene expression profiles with 20,502 features each. Quartile (RSEM) normalized gene expression data were downloaded from FireBrowse (Broad Institute, Cambridge, MA, USA) on 23 March 2019 (TCGA data version 2016_01_28). All methods were performed in accordance with their relevant guidelines; ethics approval was not required to access this publicly available dataset.

### 2.3. Data Preprocessing

Only primary tumor sample profiles (n = 497) were used. Samples that were solid tissue normal (n = 1) or metastatic (n = 52) in the TCGA-PRAD dataset were identified and removed. Data were preprocessed using an established methodology for data preprocessing and the detection of outlier and batch effects [36]. No batch effects or outliers were detected for removal.

### 2.4. Prostate Cancer Stratification into Molecular Subtypes with Pam50

The prostate adenocarcinoma samples were categorized according to the Pam50 protocol, an established protocol to determine molecular subtypes in prostate and breast cancer [37,38]. The source code was downloaded from the University of North Carolina Microarray Database [37] and was applied to assign samples to the molecular subtypes. To ensure high accuracy of class labelling, samples with confidence scores less than 0.75 (n = 157) were removed, resulting in 72 luminal A (LumA), 166 luminal B (LumB), and 102 Basal profiles.

### 2.5. Immune Profile Compilation

The existence of luminal and basal subtypes across carcinomas was previously reported [39], however, in our analysis, we only focused on the immune-function related genes. Immune-function genes (n = 923, Appendix A) that were previously identified as common immunotherapy targets [24] were extracted from the gene profiles and compiled into immune profiles for each sample. The median overall expression across the entire dataset was calculated and defined as the minimum expression threshold. To reduce computing time and increase the robustness of candidate biomarkers, genes with an expression below the minimum threshold in more than 90% of samples were considered low expressed and removed from analysis (n = 349). Finally, the filtered data were log_2_ transformed. Since the log_2_ of zero is undefined, the zero values were replaced with a computed low value of the same magnitude as the smallest non-zero value in the dataset.

### 2.6. Feature Selection of Immune-Function Genes in Prostate Adenocarcinoma

Potential biomarkers were identified for each of the comparison groups (A) LumA vs. Basal, (B) LumB vs. Basal, and (C) LumA vs. LumB as follows. Highly expressed immune-function genes (n = 574) were imported into *MFeaST*. On the comparison tab, one of the comparisons was selected (e.g., LumA vs. Basal). The *MFeaST* is an ensemble-based algorithm that relies not on one but on multiple univariable and multivariable as well as filter-, wrapper-, and embedded-type feature selection techniques. Therefore, for the identification of potential biomarkers in the current dataset, all available algorithms were selected, with 5-fold validation, optimization, and five iterations for the sequential algorithms. The 5-fold validation reduced the chance of overfitting by performing the feature selection five times, each time on a different subset of data. The ranking results of the *MFeaST* were reviewed on the Results tab, where each original feature was assigned and ranked based on an ensemble score between 0 and 1, with 1 indicating the perfect ability to discriminate between conditions of interest. The univariable feature selection methods assign a score of 0–1 to each feature. The multivariable methods search for the best combination of features that provide the highest discrimination accuracy. In the final selected set, each feature obtains a score of 1, and features that are not selected each obtains a score of 0. The results were further inspected within the software using its visualization tools to identify and select the most valuable predictors, focusing on the top-ranking 10% of the results that provided the best clustering results.

### 2.7. Differential Gene Expression of Immune-Function Genes in Prostate Adenocarcinoma

The results of the biomarkers selected with *MFeaST* were compared to the conventional approach using differential expression method. The differentially expressed results were computed as follows. Preprocessed data were imported into R version 3.6.0 (“Planting of a Tree”) for analysis with the edgeR package (version 3.25.8), as described in the user’s guide [40]. Briefly, the data were re-normalized by the counts per million (CPM) method for compatibility with edgeR, the dispersion of the data was estimated, and differential expression was calculated using the exact negative binomial test with a Benjamini-Hochberg corrected false discovery rate (FDR) [41]. All differentially expressed genes with FDR ≤ 0.05 were used in hierarchal cluster analysis and for the construction of a classifier.

### 2.8. Construction of an Immune-Function Gene Classifier

The selected biomarker candidates were further evaluated for their ability to classify the conditions of interest. Three classification analyses where performed: one using *MFeaST* selected genes, the second using differentially expressed genes (FDR ≤ 0.05), and the third using all highly expressed genes. All classification algorithms in the MATLAB Classification Learner App (n = 23) were evaluated to identify a family of classifiers that worked best for discriminating (A) LumA from Basal, (B) LumB from Basal, and (C) LumA from LumB. Preprocessed data were scaled to have 0 mean and unit-variance, then imported into Jupyter Notebook with Python 3.7 to construct final classification models with 10-fold cross-validation. For each model, the data were randomly divided into 10 disjoint sets of equal size +/−1. The training of the SVM classifier was performed 10 times and each time it was tested on a different set held out as a validation. The mean and standard deviation of accuracy, precision, recall, and F1-score were computed over the 10 validation sets to evaluate the classification model performance.

### 2.9. Clinical Information Analyses

Clinically relevant and prognostically important markers to prostate adenocarcinoma from the TCGA repository were compared between patients of different pathological types using SPSS Statistics (IBM, Armonk, NY, USA, Version 25). The Kruskal–Wallis test was used to investigate differences between variables with ordinal or ratio data. The Chi-square test was used for variables with categorical data. Results with a *p*-value ≤ 0.05 were considered significant.

### 2.10. Graphical User Interface for the Feature Selection Algorithm

A desktop application with a simple, intuitive, graphical user interface (GUI) was developed to facilitate easy application of the *MFeaST* algorithm on any dataset by end-users with different levels of computational expertise. A detailed user guide is presented in the results. While coded in MATLAB, *MFeaST* is a completely standalone application that does not require a MATLAB subscription, instead running on the freely available MATLAB Runtime software (MathWorks, Natick, MA, USA, Version 9.9). *MFeaST* can be run on most Windows or Apple desktop computers (system requirements are detailed in Appendix A and are available for download at https://www.renwicklab.com/molecular-feast/, pw: rankmolecules).

## 3. Results

### 3.1. Clinical Information Analyses

The clinicopathological statistics for important prognostic markers of prostate adenocarcinoma are presented in Table 2. No significant differences were found between the three molecular subtypes for biochemical recurrence, radiation therapy, radiation follow-up, or pN category (Chi-square test), nor were there significant differences in age or PSA levels between groups (Kruskal–Wallis test). Statistically significant differences between subtypes were found for grade (*p* = 0.020) and pathological stage (*p* = 0.009). Differences in histological type could not be assessed as one or more categories had expected counts <5. These results show that only the grade and pathological stage are the potential confounding variables in any future statistical analysis of the molecular subtypes.

### 3.2. Feature Selection of Immune-Function Genes in Prostate Adenocarcinoma

After generating immune-function profiles from TCGA-PRAD datasets, the discriminatory abilities of each immune-function gene were scored from 1 (best) to 0 (worst) using the *MFeaST* for (A) LumA from Basal, (B) LumB from Basal, and (C) LumA from LumB (Appendix A). The top-ranking 10% immune-function genes were clustered using hierarchical clustering (Figure 1). This set of genes was further reduced using the *MFeaST* visualization to the most valuable predictors for building the classification models, which resulted in (A) 33 genes used to discriminate LumA from Basal, (B) 15 genes used to discriminate LumB from Basal, and (C) 18 genes used to discriminate LumA from LumB (Appendix A).

### 3.3. Differential Gene Expression of Immune-Function Genes in Prostate Adenocarcinoma

Differential gene expression was performed to create sets of genes for classifier construction and comparison with the *MFeaST*-selected genes. After filtering for statistical significance (FDR ≤ 0.05), differential expression analyses found (A) 136 upregulated and 159 downregulated genes in LumA when compared to Basal, (B) 233 upregulated and 239 downregulated genes in LumB when compared to Basal, and (C) 161 upregulated and 167 downregulated genes in LumA when compared to LumB (Appendix A). The hierarchal clustering results for differentially expressed genes are shown in Figure 2, showing some separation, but not as distinct as in the clustering with the *MFeaST* selected genes.

The *MFeaST*-selected and differential expression selected genes are compared in Appendix A. Genes that were (1) only *MFeaST*-selected were listed under “Feature selection only”; (2) selected by both *MFeaST* and differential expression were listed under “intersection”; and (3) only differential expression selected were listed under “Differential expression only”. All but one *MFeaST*-selected gene was among the differentially expressed genes for all three comparisons (Appendix A).

### 3.4. Immune-Function Gene Classifier

When evaluated in the MATLAB Classification Learner App, linear SVM was the top performing classification algorithm in all A–C comparison cases. Therefore, all final classification models for both *MFeaST*-selected and differentially expressed gene sets were constructed using linear SVM with 10-fold cross-validation. The mean ± standard deviation of accuracy, precision, recall, and F1-score were calculated across the 10 validation folds (Table 3). Using a very small set of 15–33 features selected with *MFeaST* allowed us to construct high performing classifiers with an accuracy ranging from 81 to 95%. The precision, which is a ratio of correctly identified positive observations to the total number of positive observations (true positives + false positives) was between 80 and 97%. The recall, which measures the ratio of true positives to the total actual positives (true positives + false negatives), was between 83 and 97%. Finally, the F1-score, which is a harmonic mean of precision and recall and accounts for both false positives and false negatives, was between 79–97% for predicting the LumA, LumB, and Basal subtypes (Table 3). Comparatively, using between 295 and 472 features selected by differential gene expression resulted in less accurate classifiers with accuracy ranging from 79 to 93%, precision 78–93%, recall 78–96%, and F1-score 76–94% **(**Table 3). Although the contrast in these results between the differential gene expression results and *MFeaST* results was small, the later used a significantly smaller number of genes. For completeness, we performed the classification of the three subtypes using all 574 highly expressed immune-function genes (i.e., without feature selection or differential expression). The classification accuracy ranged from 77 to 94%, precision 74–94%, recall 75–97%, and F1-score 73–95%. This demonstrates that the *MFeaST* approach is useful for the identification of a small list of potential biomarkers, which is very important in biomarker discovery studies.

### 3.5. MFeaST Graphical User Interface

To allow all scientists, with and without computational training, the ability to select features using *MFeaST*, a simple GUI was developed. Our *MFeaST* GUI comprises six tabs that guide end-users through feature selection analysis (Figure 3). Upon opening the application, the user is presented with the “Input Data” tab (Figure 3A). Here, the user can import their data file and view a summary of the information including the filename and path, number of comparison types, number of features, number of samples, and the actual data. If there are multiple comparison types, the user can specify which binary comparisons to use in the “Comparisons Selection” table.

Next, the user can either select all or specific feature selection algorithms for analysis in the “Feature Selection” tab (Figure 3B). For small datasets, we recommend using all algorithms, as it allows for the utilization of the power of the ensemble vote; however, as computation increases proportionally with the number of features, users may consider excluding all sequential type algorithms when analyzing datasets with more than ten thousand features. As part of the biomarker discovery process, we advise removing features with low expression values or that do not vary across conditions of interest prior to the feature selection analysis. This will reduce the number of potential features and enable a more thorough analysis using the *MFeaST* approach. Additional options including the choice of cross-validation, parallel computing, and number of iterations for sequential feature selection are also provided (Figure 3B).

Once feature selection is finished, the user is presented with the “Results” tab (Figure 3C). Included in the results is a table with the option to show (1) the input data in ranked order or (2) the ranked scores for each feature, a colored scatter plot based on two selected features, and a list of final selected features (Figure 3C). The user has the option to choose all, a top percentage, or a custom list of final selected features. All ranking result tables can be exported to the Excel, CSV, or MATLAB format and the selected feature lists can be copied as plain text. Sample grouping can be visualized using the finalized list of selected features with a multidimensional visualization t-SNE technique [42]. The interface also allows users to perform hierarchical clustering analysis and generate a variety of heat maps in the “Clustering” tab using different settings (Figure 3D). Finally, the “About” tab provides information on the application.

## 4. Discussion

In biomarker discovery studies, it is often desired to identify a small set of molecular features from large -omics datasets that can be used to detect conditions of interest. However, conventional univariable approaches to identify molecular features using differential expression, do not consider the interaction between features. These approaches are confounded by the sample size, leading to the poor and sometimes misleading selection of potential biomarkers. Moreover, the final list of differentially expressed molecular features is often long, making it often difficult to select the best biomarker candidates.

In contrast, machine learning driven feature selection can be used to detect patterns and interactions in -omics datasets and lend insights into the mechanisms of the conditions studied [43]. In this manuscript, we present the *Molecular Feature Selection Tool (MFeaST)*, an ensemble feature selection desktop application that can be utilized to identify biomarkers of disease or other clinical conditions by exploring the high dimensionality of the data. To demonstrate its effectiveness, we applied *MFeaST* to prostate adenocarcinoma mRNA profiles. Using only immune-related genes, we identified a small set of features that can effectively predict three different molecular subtypes. The classification models that were built using *MFeaST*-selected genes outperformed other models built in this study using differentially expressed genes and all genes, but more importantly, used a very small set of genes.

*MFeaST* has several advantages over conventional differential expression analysis and existing feature selection methods. First, *MFeaST* is an ensemble-type method that combines multiple univariable and multivariable as well as filter-, wrapper-, and embedded-type feature selection techniques. This novel combination of algorithms makes *MFeaST* more generalizable and reliable for identifying features of importance [20]. Second, *MFeaST* utilizes the power of machine learning algorithms to rank molecular features. These algorithms characteristically rely on mathematical functions or branching logic that determine the class boundary rather than hypothesis testing, and are therefore less dependent on sample size. Using *MFeaST* allows one to select a small set of features as potential biomarkers, enabling a more effective way for biomarker discovery. From a machine learning perspective, using a small set of features can potentially avoid issues with overfitting, which is especially important for datasets with a small number of available samples. Third, we developed an intuitive, GUI that allows users, irrespective of computational expertise, to apply *MFeaST* on any dataset. Additionally, we successfully applied *MFeaST* to find biomarkers in many different omics datasets including messenger RNA-Seq [24,25,26], RT-qPCR [27,28], microRNA-Seq [29,30,31,32], NanoString profiles [33], and protein mass spectrometry [34]. These datasets were often characterized by a large number of features versus a relatively small number of samples, and usually with a lot of sparse data.

The *MFeaST* allows one to identify potential biomarkers based on a binary comparison. However, it can be extended to multiclass problems where the dataset has more than two conditions of interest. In our previous studies, we successfully applied the method in the one-vs.-all approach [24,34], where molecular features are selected by comparing one condition to all of the other conditions of interest. In addition, we utilized the *MFeaST* in a multi-layer classifier by assembling binary decision-layer models according to a predefined hierarchy [29,30,31,32]. Here, at each decision layer, the *MFeaST* was used to identify features of interests to address a binary problem. This divide and conquer approach allows one to reduce a complex problem into smaller, manageable tasks. Following this approach, a hierarchical classification of LumA, LumB, and Basal subtypes can be implemented. First the LumA vs. LumB classifier is applied on an unknown sample. If a sample is classified as LumA, then the LumA vs. Basal classifier is applied as a final prediction. On the other hand, if a sample is classified as a LumB subtype, then the LumB vs. Basal classifier is applied as a final prediction.

Using prostate adenocarcinoma mRNA data, we demonstrated how *MFeaST* allows for the selection of a small set of features for further analyses or to build classification models. Not only did the features selected with *MFeaST* result in a better classification of the three molecular subtypes than the other models in this study, but it also used a significantly smaller set of genes than in the differential expression approach. Since *MFeaST* ranks all available features—as opposed to compressing or filtering them—the final selection can be further enhanced and interpreted by a domain expert by observing each feature ranking score, assessing it visually or experimentally. Reducing a high-dimensional feature space to a small set of highly informative predictors is valuable for biomarker development and for building generalizable prediction models.

The proof-of-principle example presented here was limited by the use of the Pam50 gene set, rather than a gold-standard pathological diagnosis, to label the samples by their molecular subtype. Nevertheless, while some samples were excluded from the study due to labelling uncertainty, *MFeaST* was still able to identify a small set of features that reliably discriminated between the molecular subtypes. In addition, we did not further investigate the selected immune-related genes, highlighting the critical role that domain experts need to play in biomarker development. It is important to note that the prediction of the LumA vs. Basal subtype was relatively lower and less stable (large std) than for other comparisons across all models. This indicates that it was difficult to achieve a good and stable discrimination between the LumA and Basal samples based on the immune-function genes. Therefore, other molecular markers should be examined, and further validation of the classifier on other prostate adenocarcinoma datasets should be explored.

Here, we introduce *MFeaST* as an effective approach to identify molecular features of importance as potential biomarkers in high-dimensional -omics data. In our analyses, we focused on immune-function-related genes and investigated their potential as biomarkers using machine learning approaches. In general, *MFeaST* selected features can be further examined by domain experts through pathway analysis, literature search, and experimentation. We are continuously working on adding more functionalities to *MFeaST* and future updates may include support for multiclass feature selection and the use of unsupervised feature selection methods. We expect *MFeaST,* through its effectiveness and easy-to-use GUI, to have a large impact on existing and future -omics based biomarker discovery studies.

## Figures and Tables

**Figure 1 diagnostics-12-01997-f001:**
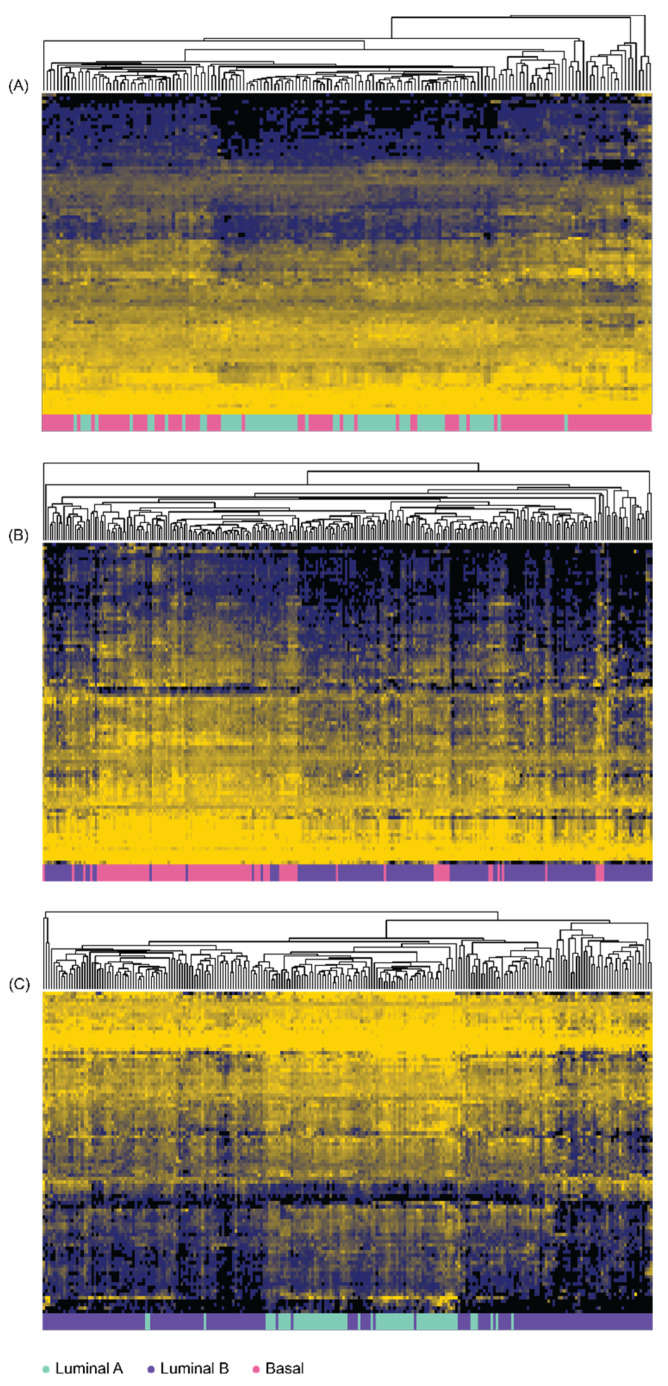
**Hierarchal clustering analysis performed using top ranked *MFeaST*-selected genes.** The top 10% highest ranking *MFeaST*-selected genes were used to cluster (**A**) Luminal A and Basal samples; (**B**) Luminal B and Basal samples; (**C**) Luminal A and Luminal B samples. Unsupervised hierarchical clustering with average linkage was performed on the log_2_ transformed expression values. The data were median-centered for proper visualization of the heatmap. Spearman correlation was used as a similarity measure between samples. The top 10% highest ranking genes are listed in Appendix A up to the horizontal double line.

**Figure 2 diagnostics-12-01997-f002:**
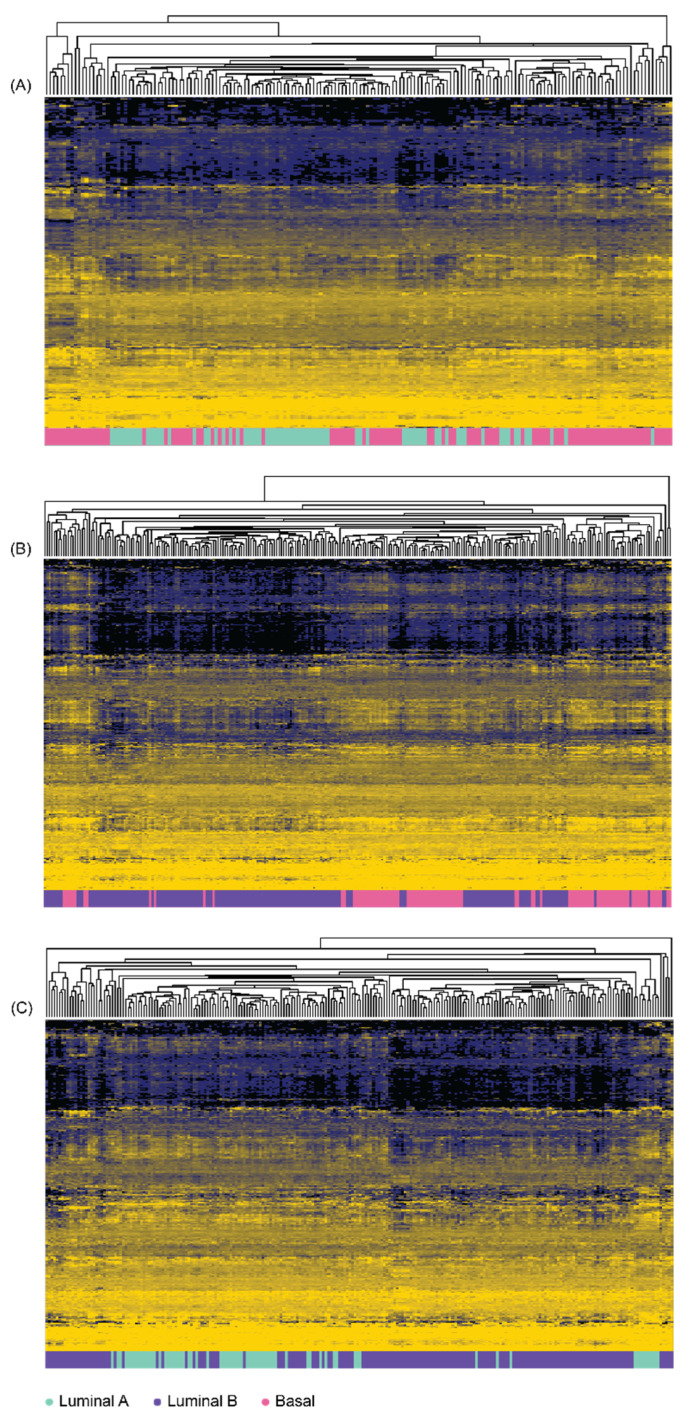
**Hierarchal clustering analysis performed using differentially expressed genes.** All differentially expressed genes (FDR ≤ 0.05) were used to cluster (**A**) Basal and Luminal A samples; (**B**) Basal and Luminal B samples; (**C**) Luminal A and Luminal B samples. Unsupervised hierarchical clustering with average linkage was performed on the log_2_ transformed expression values. The data were median-centered for proper visualization of the heatmap. Spearman correlation was used as a similarity measure between samples.

**Figure 3 diagnostics-12-01997-f003:**
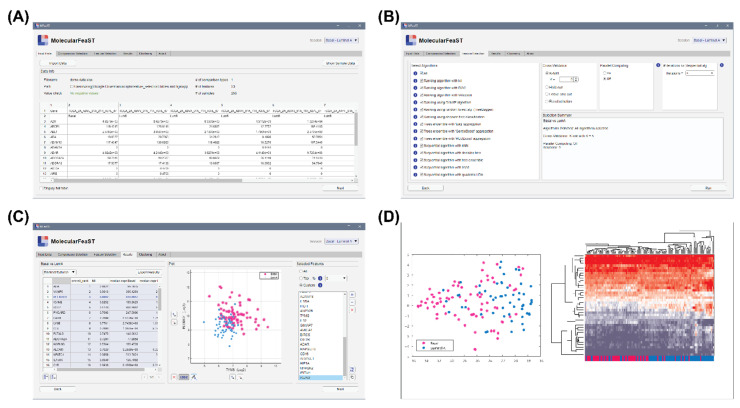
***MFeaST* application.** The *MFeaST* GUI comprises of several tabs that guide end-users through feature selection analysis. (**A**) Input data: The data can be imported and viewed. (**B**) Feature selection: The feature selection and cross validation algorithms, and number of iterations for sequential algorithm can be selected. (**C**) Results: A ranking of the features and a colored scatter plot based on two selected features are presented and can be downloaded. A final list of selected features can be created by selecting all, top percentage, or a custom list of features. (**D**) Clustering: The selected features can be visualized using t-SNE and hierarchical clustering analysis.

**Table 1 diagnostics-12-01997-t001:** The feature selection algorithms used in *MFeaST*.

Feature Selection Type	Univariable	Multivariable
Filter Type	Mutual information score ROC criteria Wilcoxon criteria ReliefF analysis	
Wrapper Type		Support vector machine K-nearest neighbors Decision tree Quadratic discriminant analysis
Embedded Type	Treebagger predictor importance	Decision tree with bagging Decision tree with gentle adaptive boosting Decision tree with random undersampling boosting

**Table 2 diagnostics-12-01997-t002:** The clinicopathologic statistics of important prognostic markers in prostate adenocarcinoma.

	Basal			Luminal A			Luminal B			Total			χ^2^	D.o.F	*p*
**Biochemical** **recurrence**	n = 88			n = 64			n = 144			n = 296					
Yes	14 (16%)			3 (5%)			24 (17%)			41 (14%)			5.77	2	0.056
No	74 (84%)			61 (95%)			120 (83%)			255 (86%)					
**Radiation** **therapy**	n = 61			n = 36			n = 99			n = 296					
Yes	8 (13%)			6 (17%)			14 (14%)			28 (9%)			0.24	2	0.888
No	53 (87%)			30 (83%)			85 (86%)			168 (57%)					
**Radiation** **follow up**	n = 80			n = 64			n = 132			n = 276					
Yes	9 (11%)			8 (13%)			25 (19%)			42 (15%)			2.76	2	0.252
No	71 (89%)			56 (88%)			107 (81%)			234 (85%)					
**Histological type**	n = 102			n = 72			n = 166			n = 340					
Acinar	101 (99%)			72 (100%)			158 (95%)			331 (97%)			* 6.10	2	0.047
Other	1 (1%)			0 (0%)			8 (5%)			9 (3%)					
**Grade group**	n = 99			n = 71			n = 164			n = 334					
GG1	10 (10%)			8 (11%)			11 (7%)			29 (9%)			21.221	6	0.020
GG2	35 (35%)			28 (39%)			31 (19%)			94 (28%)					
GG3	17 (17%)			14 (20%)			30 (18%)			61 (18%)					
GG4 + GG5	37 (37%)			21 (30%)			92 (56%)			150 (45%)					
**Pathological stage**	n = 100			n = 69			n = 166			n = 335					
pT2	41 (41%)			35 (51%)			50 (30%)			126 (38%)			9.515	2	0.009
pT3 + pT4	59 (59%)			34 (49%)			116 (70%)			209 (62%)					
**Nodal** **involvement**	n = 83			n = 62			n = 151			n = 296					
pN0	73 (88%)			52 (84%)			114 (75%)			239 (81%)			5.84	2	0.054
pN1	10 (12%)			10 (16%)			37 (25%)			57 (19%)					
	**Basal**			**Luminal A**			**Luminal B**			**Total**					
	**med**	**n**	**ran**	**med**	**n**	**ran**	**med**	**n**	**ran**	**med**	**n**	**ran**	**H**	**D.o.F**	** *p* **
Age at diagnosis	62	102	41–77	63	72	46–75	62	166	46–78	62	340	41–78	0.17	2	0.918
PSA	0.1	90	0–37.36	0.1	68	0–13.95	0.1	143	0–39.80	0.1	301	0–39.80	1.93	2	0.381

Relevant clinicopathologic data collected by the TCGA study were compared between molecular subtypes using the Chi-square or Kruskal–Wallis test where appropriate. Statistically significant differences between molecular subtypes were found for grade (*p* = 0.020) and pathological stage (*p* = 0.009). Differences in histological type could not be assessed as one or more categories had expected counts <5. Abbreviations: degrees of freedom (D.o.F), median (med), range (ran). * denote one or more cells in the category have expected counts <5.

**Table 3 diagnostics-12-01997-t003:** The classification results for the models built using the *MFeaST*-selected and differentially expressed genes.

	MFeaST	Differential Expression	All Features
(A) Basal|Luminal A			
Number of genes	33	295	574
Accuracy	81.08 ± 11.71	78.82 ± 9.77	77.12 ± 8.96
Precision	79.59 ± 19.84	78.03 ± 17.63	73.95 ± 12.80
Recall	82.86 ± 16.22	77.68 ± 13.81	75.00 ± 14.31
F1-score	78.86 ± 11.35	75.58 ± 8.80	73.09 ± 9.15
(B) Basal|Luminal B			
Number of genes	18	472	574
Accuracy	94.80 ± 2.58	92.91 ± 6.18	94.02 ± 6.61
Precision	96.56 ± 3.88	93.46 ± 7.36	93.53 ± 7.37
Recall	95.22 ± 3.75	95.74 ± 4.17	97.50 ± 4.37
F1-score	95.79 ± 2.07	94.42 ± 4.68	95.33 ± 5.03
(C) Luminal A|Luminal B			
Number of genes	15	328	574
Accuracy	95.36 ± 3.69	91.59 ± 5.20	92.45 ± 6.45
Precision	96.65 ± 5.18	93.35 ± 5.84	94.50 ± 6.36
Recall	96.99 ± 3.18	95.22 ± 6.09	95.26 ± 6.08
F1-score	96.70 ± 2.56	94.05 ± 3.68	94.66 ± 4.36

## Data Availability

Not applicable.

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
