# Peer review of "A Computational Approach to Identification of Candidate Biomarkers in High-Dimensional Molecular Data"

_diagnostics, 2022, doi:10.3390/diagnostics12081997_

Round 1

Reviewer 1 Report

Review : “A novel Computational approach to identification of  candidate biomarkers in high dimensional molecular data” by Gerolami at al

In this manuscript authors introduce a new approach to select “biomarkers” from a set of molecular features that are able to make prediction in a binary classification problem.

Questions:

What is the goal of the biomarker selection?

-          -Is it to make the best predictions for the data available? This is tested, but you already know their class.

-          -Is it to detect features that can be used to predict status for a new sample? This is never tested as all samples in the data were used to make preprocessing decisions and also all samples were used to select features. The final cross-validation cannot be used to test the selected features. A proper validation should include samples from a real test set, meaning that test samples have not been used in any way in the model building. This can be applied using a double cross validation procedure of a real test set. [In section 2.6 authors mention that 5 fold validation was used for identification of potential biomarkers. Perhaps they mean that this is their double cross validation approach, but this was not clearly described.] Furthermore, samples from a different study on prostate adenocarcinoma subtypes will be slightly different to the set you have been using. This would be the best approach of validating the usefulness of the selected features.

-          - Is it to find features that could be tested in new experiments? In that case a false discovery rate indication should be available.

Authors discuss that their approach is develop to prevent overfit. As overfit is a problem when small sample sizes are used, the H0 situation should be tested. In an H0 situation (meaning that there is no difference between the groups), how does the biomarker selection approach behave? Does it select features that are able to make class predictions? This should be tested and discussed.

The user is allowed to set weights to give some feature selection methods more ranking power over others. Based on what information should the user decide how to set the weights?

For which types of data would the methods be useful? Can it deal with the zero problem in e.g. microbiome data or with detection limit issues in metabolomics and proteomics data?

You showed the method in binary classification in a multiclass problem. How will you make a prediction for a new sample where you do not know which class it belongs to?

10% of the features are selected: What about false discovery rates? Can false discovery rates be calculated for the selected features and can it be used to decide on how many features will be selected?

The ensemble score provides information on discrimination ability based on ensemble score, where 1 indicates perfect ability to discriminate between conditions. How is this ensemble score calculated for multivariate models where interactions between features are used to make predictions?

Clustering is applied on the log transformed median centered expression values. Was the data also preprocessed in this way before feature selection? Similarities between samples (Spearman) and genes (Euclidean) is calculated, but for only one of them (not clear which) this is indicated with a dendrogram.

Classification results (Table 3) show large standard deviations for Basal Luminal A classification results. How is this possible? How are the stds calculated? Is this over the 5 fold double crossvalidation or over the 10 fold single crossvalidation (with the same set of genes)? Please clarify.

Author Response

Response to Reviewer 1 Comments

Point 0:  “A novel Computational approach to identification of candidate biomarkers in high dimensional molecular data” by Gerolami at al

In this manuscript authors introduce a new approach to select “biomarkers” from a set of molecular features that are able to make prediction in a binary classification problem. 1).

Thank you very much for reviewing our work.

Point 1: What is the goal of the biomarker selection?

Is it to make the best predictions for the data available? This is tested, but you already know their class.

Is it to detect features that can be used to predict status for a new sample? This is never tested as all samples in the data were used to make preprocessing decisions and also all samples were used to select features. The final cross-validation cannot be used to test the selected features. A proper validation should include samples from a real test set, meaning that test samples have not been used in any way in the model building. This can be applied using a double cross validation procedure of a real test set. [In section 2.6 authors mention that 5 fold validation was used for identification of potential biomarkers. Perhaps they mean that this is their double cross validation approach, but this was not clearly described.] Furthermore, samples from a different study on prostate adenocarcinoma subtypes will be slightly different to the set you have been using. This would be the best approach of validating the usefulness of the selected features.

 Is it to find features that could be tested in new experiments? In that case a false discovery rate indication should be available.

Response 1: Thank you for identifying this clarification point. We agree that our introduction of feature selection needed improvement. As the reviewer correctly pointed out, the main goal of the feature selection is to reduce the number of features to a small set of most variable predictors required for classification. This approach is also ideal for identifying potential biomarkers and/or biologically relevant molecules in large datasets.

To address this point we reworked the second paragraph of the introduction to provide a better background into machine learning-based feature selection (lines 45 – 58)

Feature selection is a machine learning technique for identifying features of importance in high dimensional data. The main goal of the feature selection is to reduce the number of features to a small set of most variable predictors required for classification. A typical  –omics dataset has a relatively low sample size compared to the number of features. This affects classifier accuracy and increases the risk of overfitting the prediction model by developing an overly complex classifier with perfect classification on a training set, but low performance and generalizability on unseen data [6-13]. Therefore, when developing generalizable prediction models from -omics data, it is favourable to use only the most important predictors. Feature selection is a supervised learning approach that evaluates features either individually (univariable approach) or as a group (multivariable approach). Feature selection algorithms are typically based on (i) filter methods that evaluate each feature without any learning involved; (ii) wrapper methods that use machine learning techniques for identifying features of importance; or (iii) embedded methods where the feature selection is embedded with the classifier construction [6].

We agree that with respect to the prediction of Luminal A, Luminal B and Basal subtypes in adenocarcinoma, it is best to perform a hold out validation. The main objective of the manuscript was to demonstrate the effectiveness of the MFeaST to identify a small set of most discriminating features. We used the classification of the molecular subtypes in prostate adenocarcinoma as a proof-of-principle example for MFeaST. Due to the relatively small number of samples available for this study, hold out validation was not possible. We believe that for this dataset 10 fold cross-validation was the most appropriate method to demonstrate effectiveness of MFeaST to identify a small set of important molecular features. We added in the discussion the acknowledgement that further external validation of the molecular subtypes is required for the study of prostate adenocarcinoma on lines 408-413

Therefore, other molecular markers should be examined and further validation of the classifier on other prostate adenocarcinoma datasets should be explored. 

We also explain our cross-validation approach and address manuscript revisions in Response 2.

Since MFeaST ranks all available features - as opposed to compressing or filtering them - the final selection can be further enhanced and interpreted by a domain expert by observing each feature ranking score, assessing it visually or experimentally. We explain this point in more detail and address manuscript revisions in Response 6.

Point 2: Authors discuss that their approach is develop to prevent overfit. As overfit is a problem when small sample sizes are used, the H0 situation should be tested. In an H0 situation (meaning that there is no difference between the groups), how does the biomarker selection approach behave? Does it select features that are able to make class predictions? This should be tested and discussed.

Response 2: We apologize for the lack of clarity. In the context of supervised learning and classification, the overfitting refers to developing an overly complex classifier with a complex decision boundary (e.g. a multivariable polynomial). While this classifier allows perfect classification on the training set, it is unlikely to perform well on a new, unseen data (Duda, R.O.; Hart, P.E.; Stork, D.G., Pattern Classification, 2 ed. New York: Wiley, 2001). One effective way to avoid overfitting is by reducing the number of features used in the classification model.  This is achieved with the feature selection method.

The MFeaST ranks each feature based on how well it individually or in combination with other features discriminates conditions of interest. Subsequently, the MFeaST provides the functionality to select the features that contribute to the best class prediction from the ranked list of features. In the manuscript we show an approach to reduce the large number of features to a small set of most discriminating features, and therefore, lessen the chance of overfitting.  

A common approach for detecting overfitting is by using cross-validation. In this study we used a 10-fold cross-validation to estimate the generalizability of the SVM classifiers in each classification of A) LumA from Basal, (B) LumB from Basal, and (C) LumA from LumB. For each model the data were randomly divided into 10 disjoint sets of equal size. The training of the SVM classifier was performed 10 times and each time it was tested on a different set held out as a validation set. In addition, extra consideration to avoid overfitting was taken with the feature selection, prior to the classification. The MFeaST was also applied with a 5-fold approach, where all feature selection models were executed 5 times, each time using a different subset of data. The final classification results presented in Table 3 are the average across the 10 validation sets.  Since the mean accuracy, precision, recall, and F1-Score for the testing folds are high, the possibility of overfitting is low.

We thank the reviewer for identifying these points and we agree that they require more clarity. We added clarification regarding overfitting to the introduction (please see response 1), we added an explanation of the 5-fold validation of the feature selection on lines 145-149:

Therefore, for identification of potential biomarkers in the current dataset all available algorithms were selected, with 5-fold validation, optimization, and 5 iterations for the sequential algorithms. The 5-fold validation reduced the chance of overfitting by performing the feature selection 5 times, each time on a different subset of data.

and we clarified the 10-fold validation of the classifiers on lines 178 – 182 of the revised manuscript :

For each model the data were randomly divided into 10 disjoint sets of equal size +/-1. The training of the SVM classifier was performed 10 times and each time it was tested on a different set held out as a validation. Mean and standard deviation of accuracy, precision, recall, and F1-score were computed over the 10 validation sets to evaluate classification model performance.

Point 3: The user is allowed to set weights to give some feature selection methods more ranking power over others. Based on what information should the user decide how to set the weights?

Response 3: We thank the reviewer for noting the missed information in the manuscript. The greedy algorithm that assembles the results of multiple feature selection methods applies weights to weight each feature selection method differently. The decision for setting the weights depends on testing which family of classifiers works best for a given dataset. Although it is part of the greedy algorithm and our feature selection algorithm, this functionality is not fully utilized in the MFeaST graphical user interface (GUI) that accompanies the manuscript. In the current GUI implementation all feature selection methods are set to have the same weight. To avoid any further confusion to the readers we removed the reference to the weights from the manuscript.

Point 4: For which types of data would the methods be useful? Can it deal with the zero problem in e.g. microbiome data or with detection limit issues in metabolomics and proteomics data?

Response 4: Thank you for the very helpful comment. Yes, our approach can deal with sparse data that has many zero values. In our preprocessing, we log2 transformed the data. Since logarithm of zero is undefined, the zero values were replaced with a low value that is 10 to the power of the (exponent-1) of the lowest non-zero value in the data. For example, if the minimum non-zero value in the dataset is 0.089, then all the zero values are replaced with 0.001.  In addition, we filtered out features with low expression across all samples, which reduced the data sparsity as well as dimensionality of the data.

We realize that we inadequately explained this preprocessing step in the manuscript and included the log2 transformation method on lines 135-137:

Finally, the filtered data were log2 transformed. Since log2 of zero is undefined, the zero values are replaced with a computed low value of the same magnitude as the smallest non-zero value in the dataset.

The MFeaST works well on many different types of data. To date we applied MFeaST to identify features of interest in RNA-seq, RT-qPCR, microRNA-seq, NanoString profiles, and protein mass spectrometry.  We cited these studies in the discussion, on lines 372 – 375. A recent publication was also added to the list, where the MFeaST was applied in study of sex differences in the aging murine urinary bladder. In addition, we added the emphasis that the application was applied on data with high dimensionality versus a relatively small sample size with sparse data on lines 375 – 377:

These datasets were often characterized by a large number of features versus relatively small number of samples and usually with a lot of sparse data.

We also tested MFeaST on imaging data as part of a patent application on the classification of neuroendocrine neoplasms (US Provisional Patent Application No. 63/052,267) that heavily utilized the MFeaST.  It was able to identify features that provided higher classification accuracy than without the feature selection. We also successfully applied the MFeaST on several microbiome datasets as part of an undergraduate thesis to predict recurrence of Clostridium difficile infection. Recently, the MFeaST was applied to rank a very sparse ChipSeq data to identify genomic locations with most discriminating peaks (manuscript in preparation). However, since these studies are not published, we did not mention them in the manuscript.

Point 5: You showed the method in binary classification in a multiclass problem. How will you make a prediction for a new sample where you do not know which class it belongs to?

Response 5: Thank you for raising this important question. To address the multiclass problem we developed and applied a hierarchical classification approach in several of our classification studies (ref. 29, 31 and 32). Following this approach, a hierarchical classification of Luminal A, Luminal B and Basal subtypes can be implemented. First the Luminal A vs Luminal B classifier is applied on an unknown sample. If a sample is classified as Luminal A, then the Luminal A vs Basal classifier is applied as a final prediction. On the other hand, if a sample is classified as a Luminal B subtype, then the Luminal B vs Basal classifier is applied as a final prediction.

We discuss our hierarchical classification approach in the discussion and we added the exact answer above as an example of how an unknown sample can be classified as Luminal A, Luminal B or Basal subtype on lines 386 – 391.

Point 6: 10% of the features are selected: What about false discovery rates? Can false discovery rates be calculated for the selected features and can it be used to decide on how many features will be selected?

Response 6: We apologize for the lack of clarity. The feature selection described in the manuscript is based on the machine learning approach. This is a supervised learning technique where mathematical functions, decision boundaries and/or heuristics that minimize the classification error between conditions of interest are determined. The features are ranked based on how well the identified function (or a decision boundary, etc) separates the conditions of interest. Starting at 1% of the resulting ranked features the investigator inspects how the selected features separate the conditions of interest. In the manuscript we identified that using the top10% of the top ranking features provided the best clustering results. The user can further investigate each feature individually or as a group through visualization to further narrow down the set of selected features. We added this clarification to the revised manuscript on

lines 152-158:  The results were further inspected within the software using its visualization tools to identify and select the most valuable predictors, focusing on the top-ranking 10% of the results which provided the best clustering results.

and lines 331-333: All ranking result tables can be exported to Excel, CSV or MATLAB format and the selected feature lists can be copied as plain text.

The feature selection approach does not use hypothesis testing, and therefore, the final selection of features does not rely on a p-value or on the FDR-corrected p-values.  Each feature selection algorithm provides a score between 0 and 1. A user can review and download the assembled and sorted scores for an investigation and use them as another piece of information in the feature selection decision. In the manuscript we compared our feature selection method to the differential expression analysis, where FDR values were calculated and used to identify differentially expressed genes.

Point 7: The ensemble score provides information on discrimination ability based on ensemble score, where 1 indicates perfect ability to discriminate between conditions. How is this ensemble score calculated for multivariate models where interactions between features are used to make predictions?

Response 7: Thank you for identifying this clarification point. The multivariable feature selection methods search for the best combination of features that provide the highest discrimination accuracy. In the final selected set each feature gets a score of 1 and features that are not selected each gets a score of 0. We clarified this point on lines 152-155 of the revised manuscript:

The univariable feature selection methods assign a score of 0-1 to each feature. The multivariable methods search for the best combination of features that provide the highest discrimination accuracy. In the final selected set each feature gets a score of 1 and features that are not selected each gets a score of 0.

Point 8: Clustering is applied on the log transformed median centered expression values. Was the data also preprocessed in this way before feature selection? Similarities between samples (Spearman) and genes (Euclidean) is calculated, but for only one of them (not clear which) this is indicated with a dendrogram.

Response 8: Thank you for noticing these discrepancies in our description. As a last step of our data preprocessing the data were log2 transformed. Please see Response 4 for addressing this issue.

The median centering was only applied to clustering for proper visualization of the heatmap. The median centering is performed by subtracting the overall median from all expression values and therefore it does not alter the data distribution, only shifts the median towards zero. This is necessary for the proper colouring of the heatmap.

We added the clarification to the figure caption of Figures 1 and 2.

The clustergram allows to apply hierarchical clustering to both rows and columns of the data using different similarity measures. For clarity of the image, we removed the dendrogram for the genes as that didn’t provide any useful information and only presented the dendrogram for the samples.  However, we agree that the figure caption creates confusion, and therefore, we removed the reference to the similarity measure of the genes from the figure captions.

Point 9: Classification results (Table 3) show large standard deviations for Basal Luminal A classification results. How is this possible? How are the stds calculated? Is this over the 5 fold double crossvalidation or over the 10 fold single crossvalidation (with the same set of genes)? Please clarify.

Response 9: Thank you for raising this point. The standard deviations in Table 3 are calculated based on the 10-fold cross validation. We added the clarification on lines 271-273 of the revised manuscript:

The mean ± standard deviation of accuracy, precision, recall, and F1-score were calculated across the 10 validation folds (Table 3).

The large standard deviations indicate that the classification accuracy significantly varies between the validation folds. The large deviations are also present for the classification using the differentially expressed genes and using all highly expressed genes. The overall Accuracy, Precision, Recall and F1-scores are also lower compared to other classification models. This implies that it is difficult to find a good and stable discrimination between Basal and Luminal A samples based on the immune-function genes. The Luminal A, LuminalB and Basal subtypes are not well studied in prostate adenocarcinoma, however it has been shown that Luminal B prostate cancers exhibited the poorest clinical prognosis (PMID: 28494073). Therefore, it can be hypothesised that Luminal B is more different than Lumina A and Basal subtypes and therefore easier to discriminate.  Further analysis of subtypes and other genomic features is warranted.  This is important point and we added the limitation in the discussion, lines 408 – 413:

It is important to note, that the prediction of LumA vs Basal subtype was relatively lower and less stable (large std) than for other comparisons across all models. This indicates that it was difficult to achieve a good and stable discrimination between LumA and Basal samples based on the immune-function genes. Therefore, other molecular markers should be examined and further validation of the classifier on other prostate adenocarcinoma datasets should be explored.

Reviewer 2 Report

The author established the Molecular Feature Selection Tool (MFeaST) that enables identification of candidate biomarkers in -omics data by reduction of features to the most valuable predictors. This novel, ensemble-type feature selection technique does not rely just on one approach but utilizes multiple univariable and multivariable as well as filter-, wrapper- and embedded-type feature selection techniques. Albeit, this analysis shows that the biomarkers selected using MFeaST can also be used to build accurate classification models that perform better in all comparisons than a model built using differentially expressed genes or without feature selection, I still have some suggestions.

1, All figures are highly professional, and the authors should guide the readers to the meaning of the images appropriately; otherwise, it is likely to cause misunderstandings. Therefore, I suggest that the author consider revising these figure legends again.

2, In figure 1 and 2, it would be much better if the author can label important genes in the heatmap.

3, There are a few typo issues for the authors to pay attention. Please unify the writing of scientific terms. “Italic, capital” ? make it consistent throughout the whole manuscript. For example, Page 5, Line 189” chi square test and Kruskal-Wallis test ”, which needs be corrected as “chi-square” 

Author Response

Response to Reviewer 2 Comments

Point 0:  The author established the Molecular Feature Selection Tool (MFeaST) that enables identification of candidate biomarkers in -omics data by reduction of features to the most valuable predictors. This novel, ensemble-type feature selection technique does not rely just on one approach but utilizes multiple univariable and multivariable as well as filter-, wrapper- and embedded-type feature selection techniques. Albeit, this analysis shows that the biomarkers selected using MFeaST can also be used to build accurate classification models that perform better in all comparisons than a model built using differentially expressed genes or without feature selection, I still have some suggestions.

Thank you very much for reviewing our work.

Point 1: All figures are highly professional, and the authors should guide the readers to the meaning of the images appropriately; otherwise, it is likely to cause misunderstandings. Therefore, I suggest that the author consider revising these figure legends again.

Response 1: Thank you for noting the lack of clarity of the figure legends. These were revised to emphasize the differences between the figures and methods were cleaned up to avoid misunderstandings.

Figure 1. Hierarchal clustering analysis performed using top ranked MFeaST-selected genes. The top 10% highest ranking MFeaST-selected genes were used to cluster A) Luminal A and Basal samples; B) Luminal B and Basal samples; C) Luminal A and Luminal B samples. Unsupervised hierarchical clustering with average linkage was performed on log2 transformed expression values. The data were median-centered for proper visualization of the heatmap. Spearman correlation was used as a similarity measure between samples. The top 10% highest ranking genes are listed in the Supplementary table 3 down to the horizontal double line.

Figure 2. Hierarchal clustering analysis performed using differentially expressed genes. All differentially expressed genes (FDR ≤ 0.05) were used to cluster A) Basal and Luminal A samples; B) Basal and Luminal B samples; and C) Luminal A and Luminal B samples. Unsupervised hierarchical clustering with average linkage was performed on log2 transformed expression values. The data were median-centered for proper visualization of the heatmap. Spearman correlation was used as a similarity measure between samples.

Figure 3. MFeaST application. The MFeaST GUI comprises of several tabs that guide end-users through feature selection analysis. A) Input data: The data can be imported and viewed. B) Feature Selection: The feature selection and cross validation algorithms, and number of iterations for sequential algorithm can be selected. C) Results: A ranking of the features and a colored scatter plot based on two selected features are presented and can be downloaded. A final list of selected features can be created by selecting all, top percentage, or a custom list of features. D) Clustering: The selected features can be visualized using t-SNE and hierarchical clustering analysis.

Point 2: In figure 1 and 2, it would be much better if the author can label important genes in the heatmap.

Response 2: Thank you for your suggestion. There are 92-477 genes used to generate these figures and labeling these would clutter the diagram, making them not legible. Therefore, gene labels were omitted in the figures. However, we added a double line to Supplementary Table 3 to indicate the extent of the top 10% highest ranking genes that were used to generate Figure 1. We also noted the double line indication in the revised figure caption. Figure 2 shows differentially expressed genes with FDR <= .05 and these are listed in the Supplementary table 4.

Point 3: There are a few typo issues for the authors to pay attention. Please unify the writing of scientific terms. “Italic, capital” ? make it consistent throughout the whole manuscript. For example, Page 5, Line 189” chi square test and Kruskal-Wallis test ”, which needs be corrected as “chi-square” 

Response 3: Thank you for noting the inconsistencies in the style of the scientific terms. These were corrected throughout the manuscript. The chi-square was corrected to Chi-square on lines 193 and 204 of the revised manuscript. The following inconsistencies in the style were addressed:

  • Hold out -> hold-out on page 2
  • Leave one out -> leave-one-out on page 2
  • luminal and basal on page 3
  • counts per million -> Counts per Million on page 4
  • TCGA PRAD -> TCGA-PRAD on page 7
  • F1-Score -> F1-score on pages 11 and 12
  • machine-learning -> machine learning on page 14
  • ensemble type -> ensemble-type on page 14
  • multi-class -> multiclass on page 15